# Low Power Environmental Image Sensors for Remote Photogrammetry

**DOI:** 10.3390/s22197617

**Published:** 2022-10-08

**Authors:** Alpha Yaya Balde, Emmanuel Bergeret, Denis Cajal, Jean-Pierre Toumazet

**Affiliations:** 1Laboratoire de Physique de Clermont-UMR 6533 (LPC), Centre National de la Recherche Scientifique (CNRS), Université Clermont Auvergne (UCA), Avenue Aristide Briand CS 82235, CEDEX, 03101 Montluçon, France; 2Laboratoire de Géographie Physique et Environnementale (GEOLAB), Centre National de la Recherche Scientifique (CNRS), Université Clermont Auvergne (UCA), 63000 Clermont-Ferrand, France; 3LTSER, Zone Atelier Loire, UMR 7324—CITERES, MSH Villes et Territoires BP 60449, CEDEX 03, 37204 Tours, France

**Keywords:** energy consumption, photogrammetry, IoT, image sensor network, energy optimisation, 3D reconstruction, environment monitoring

## Abstract

This paper aims to prove the feasibility of a 4D monitoring solution (3D modeling and temporal monitoring) for the sandbar and to characterize the species’ role in the landscape. The developed solution allows studying the interaction between the river dynamics and vegetation using a network of low resolution and low power sensors. The issues addressed concern the feasibility of implementing a photogrammetry solution using low-resolution sensors as well as the choice of the appropriate sensor and its testing according to different configurations (image capture and storage on the sensor and/or image transmission to a centralization node) and also the detailed analysis of the different phases of the process (camera initialization, image capture, network transmission and selection of the most appropriate standby mode). We reveal that the tiny, low-cost board (ESP32-Cam) can perform a 3D reconstruction and propose using the camera’s UXGA (1600, 1200) resolution because of the quality rendering and energy consumption. A multi-node scenario based on a combined Wi-Fi and GSM relay is proposed in the study showing several years of autonomy for the system. Finally, to illustrate the energy cost of the module, we have defined a study process, where we have identified and quantified one by one the different phases of operation of the card for better energy optimization (setup, camera configuration, shooting, saving on SD card, or sending by Wi-Fi). The device is now operational for deployment on the Allier River (France).

## 1. Context

From an application point of view, 4D monitoring (3-D modeling + temporal monitoring) remains relevant in several fields such as collecting information on forests [1], studying the dynamic characteristics of the structure of a bridge [2], and monitoring the interaction between the dynamics of a river and the vegetation. This paper aims to present the energy balance of a system that will be deployed for remote monitoring of a study area on the Allier River, one of the last largest wild rivers in Europe [3,4]. The objective is to better understand the main phenomena related to the dynamics of the river and that can influence it: bank erosion, sediment deposition, the interaction of the river with the riparian vegetation, etc. This analysis requires shooting images in places that are difficult to access without the possibility of having any energy source. The presented study aims to prove the feasibility of this approach. This report focuses on a context allowing the recovery of photographs under the necessary conditions for reconstruction by photogrammetry. This method requires many shots from different angles and is usually performed by an operator from photos taken with a standard camera while moving around the area of interest: this is the Structure from Motion (SFM) approach. Recently, in order to simplify and automate these shots, the use of drones has been introduced [5,6]. However, like the classical approach, it requires the presence of an operator on site and thus does not allow systematic monitoring of preeminent events. Therefore, we propose to deploy a network of low-resolution, energy-efficient, and communication-capable image sensors, allowing remote transmission of images for further processing. These sensors can either transmit images regularly or save them on microSD.

## 2. Introduction

Continuous sensing [7], has become a new paradigm due to real-time collection and measurement, it provides the opportunity to use advanced technologies in the human environment. The Internet of Things introduces a new way [8] of doing photogrammetry because it goes beyond the limits of traditional methods. In this study, we use connected devices to monitor the dynamics and interaction of the Allier river with the surrounding vegetation. The work presented in this article aims to design a network of communicating sensors to monitor the bed and banks of a river. The idea is to be able to follow the river by photogrammetry, i.e., the banks (sand, gravel) and the associated vegetation, which takes advantage of this substrate to settle and contributes to stabilizing it by its root network. Photogrammetry is a technique that followed the invention of photography, previously oriented towards the realization of land maps (topography). With the new technologies, its field of intervention has widened, and it helped to have another perceptive on the evolution of the environment, for example, the interest that can have the deposits of sand dunes in the proliferation of grass on the banks. Traditionally digital cameras and, recently, drones are the tools used for the various captures in the framework of a 3D reconstruction. Several studies of photogrammetry using drones have already been carried out: [4,6,9], for which the 3D monitoring was conducted using an SFM (Structure From Motion) photogrammetry technique with aerial images (aircraft + drone). This analysis is efficient, but it allows only a posteriori vision of the effects of a flood or an intense drought, the overflights being carried out with a periodicity of several months. Only the global result is accessible without information on the different steps that led to this result. The proposed approach ensures continuous monitoring, allowing explanation of causes and locating in time the beginning and the end of a phenomenon, thanks to the collected data. The energy balance of this type of system, which would permit us to estimate the operating time of this system and the means to optimize it, is not well documented in the literature. The approach presented in the rest of the article consisted in validating the feasibility of a photogrammetric reconstruction based on low-resolution sensors. Configurations for better coverage of the area are explained. After that, identifying the energy consumption of each of the steps of the image-taking (sensor configuration, image taking, image transmission, and standby) is conducted to quantify the energy consumed and finally define an optimized scenario allowing long-term monitoring with low energy consumption.

## 3. Photogrammetry and Associated Parameters

The principle of Structure From Motion (SFM) photogrammetry is usually to make, with the same sensor, a series of images moving along or around the element to be modelled, depending on the configuration of the study object. The advantage of this technique is that it does not require knowing the position of each shot to reconstruct the scene or object in 3D: it is the triangulation effect obtained by detecting several common points on a succession of different images that allows locating both these points in space but also the position of each shot. Once this first phase is completed, dense point clouds are determined, looking for a maximum of common points between the images. According to this principle, we propose an approach that consists in positioning a set of sensors and fixing them in such a way so as to carry out a reliable and, above all, reproducible follow-up of the temporal evolution of the studied area from an always identical device. In this context, optimizing the number of sensors required is essential. The constraint is that to allow the triangulation necessary for 3D reconstruction, each pixel representing a point of interest must be present on at least three different images. In concrete terms, this means that two consecutive images must overlap at least 66% (Figure 1).

Figure 1 defines the different geometrical parameters to be taken into account to optimize the system: a series of sensors (Si) is located at a distance (*d*) from the study area. For each sensor, the field of view (*FOV*) is expressed as a function of the distance (*d*) and the angle of view (θ) by the relation: (1)$$FOV=2*d*\tan\frac{\theta }{2}$$

In order to allow an overlap of 66% of the images, the distance between 2 sensors, noted $S_{1}S_{2}$ in Figure 1, must be at most: (2)$$S_{1}S_{2}=\frac{FOV}{3}=\frac{2*d*\tan\frac{\theta }{2}}{3}$$

In this case, the area covered for photogrammetric analysis for n sensors deployed will be: (3)$$Coverage=\frac{\left(n-2\right)*FOV}{3}=\left(n-2\right)*\frac{2*d*\tan\frac{\theta }{2}}{3}$$

Knowing the size of the area to be studied and the distance at which it is located, it is, therefore, possible to determine the number of sensors needed. Of course, the final resolution, i.e., the area on the ground represented by a pixel, will depend on the resolution of the chosen sensor: this is what we will now discuss.

## 4. Influence of the Resolution and Positioning of the Sensors

The purpose of this section is to review the different geometric parameters that contribute to a better coverage of an area. The resolution and the positioning of the sensors are two adjustable parameters.

### 4.1. Influence of the Resolution

As previously mentioned, photogrammetry is usually performed with high-resolution cameras. Before implementing the presented device, it was, therefore, essential to test the feasibility of using low-resolution sensors for photogrammetry. The method used to carry out this test consisted in starting from high-resolution images obtained from sensors of reflex type and then degrading these images to assess the influence of this reduction on the quality of the photogrammetric rendering model. With a fixed distance, we have in Table 1 the results of the process described above.

We can see that the quality of the 3D model, represented here by the number of points, decreases gradually with the size of the image [10], which is logical. On the other hand, we note that below a size of 400 × 300 pixels, the photogrammetric reconstruction is no longer possible under our test conditions. There is, therefore, in absolute terms, a minimum image size below which it is impossible to go. More than the size of the image in pixels, it is its weight (in kilobytes) that will be the most determining parameter because it directly impacts the duration of the transmission, and observation felt in the energy consumed. The degradation mechanism illustrated in Table 1 shows that the resolution of the 3D model does not vary linearly with the image’s resolution. We can notice in particular that the number of points is reduced by half (43%) between 3200 × 2400 (8 Mpx) and 2400 × 1800 (4 Mpx); the same observation (54%) is made for 1600 × 1200 (2 Mpx) and 1200 × 900 (1 Mpx). Among the different resolutions presented, the appropriate choice is between 3200 × 2400 and 1600 × 1200. The market price of the cameras depends on the manufacturer [11] and the resolution. Concerning the manufacturer Omnivision, the active power requirements for its model Ov8850 (8 Mpx) [12] is four times larger than that of Ov2640 (2 Mpx) [13]. This observation led us to prefer the resolution 1600 × 1200, corresponding to the UXGA format, which appears to be a good compromise. Figure 2 and Figure 3, showing a 3D rendering obtained with agisoft metashape software, illustrates the difference between the two resolutions mentioned above.

The resulting model contains white areas, corresponding either to areas occupied by water or areas of sky, for which the photogrammetric model cannot identify a correspondence between the different images. This simple visual comparison allows us to verify the possibility of making relevant 3D models but not to quantify the real performance of the model in terms of resolution because it depends on the distance at which the images are taken. Indeed, more than the total number of points of the model, it is the number of points per surface unit that will be determined to define the resolution of our device and validate its capacity to analyze the interactions between the vegetation and the morphology of the river finely.

### 4.2. Influence of the Distance

We have chosen to implement the ESP32-Cam image sensor, a device widely used in IoT applications [14,15,16] compatible with the camera model Ov2640. The ESP32 module is a microcontroller from Espressif and Ai-Thinker, with advanced features such as Bluetooth, Wi-Fi, and an SD card. This tiny board (27 mm × 40.5 mm) includes a microcontroller with embedded memory (448 kB ROM, 520 kB SRAM) and a socket for a microSD card. It is associated with a mini camera (Ov2640), this is why it is called ESP32-Cam. It is a low power consumption module, therefore ideal for IoT projects. To evaluate the performance of this sensor in our field of application, we have conducted a series of tests by applying photogrammetry on a reduced area of 31 m long and representing an area of 93 m${}^{2}$ measured in the experimental zone.

We have performed two tests, one at a distance of 6 m and the other at 12 m. With the ESP32-Cam, the angle of view is 53°, and the *FOV* is therefore equal to the distance *d*. Under these conditions, the relationship (2) becomes: (4)$$S_{1}S_{2}=\frac{FOV}{3}=\frac{d}{3}$$

The minimum distance between two successive sensors is 2 m if they are placed at a 6 m distance and 4 m if they are placed at 12 m. Under these conditions, the minimal number of sensors to be implemented to cover the area of 30 m length is, therefore, according to Equation (Equation 3): (5)$$n=\frac{3*coverage}{FOV}+2$$

In our case, we need a minimum of 17 sensors at a distance of 6 m and 10 at a distance of 12 m. Before validating the dimensioning of our device and testing the interest of an over dimensioning of the device, we placed a sensors network at a 6 m distance from the study area, and we varied the distance between the sensors from 2 m (previously calculated value) to 0.5 m, by steps of 0.5 m. We repeated the same experiment, this time at a distance of 12 m, and varied the distance between the sensors from 4 m to 1 m, in steps of 1 m. Table 2 and Table 3 present the results obtained:

These tests allow us to conclude on several points. First, they validate the calculation of the minimum number of sensors allowing the realization of the photogrammetric model. Under these conditions, at a distance of 6 m, we obtain a density of 13,154 points per m${}^{2}$, that is to say, approximately 1 point per cm${}^{2}$, which makes it possible to describe topo-environmental variations with a very high degree of precision. At a distance of 12 m, the resolution decreases by 2763 points per m${}^{2}$ or about 1 point per 5 cm${}^{2}$. These resolutions are still very high compared to those obtained with more traditional methods, such as airborne LiDAR or aerial photogrammetry, which provide a maximum resolution of about 4 points per m${}^{2}$ [17] and confirm the ability of our device to analyze acceptable variations, but at a tiny scale. At the same time, LiDAR surveys allow only a macroscopic analysis but at a much larger scale. However, it should be noted that the resolution of our system decreases rapidly with distance. Depending on the constraints of the terrain, it will be necessary to be vigilant to this parameter, which will potentially significantly influence our results’ accuracy. As previously indicated, the purpose of our system is to focus on close monitoring of small areas of interest, which is validated by these first results. We also observe that adding image sensors regularly interspersed between the previous ones significantly affects the final resolution of the 3D model. Thus, by multiplying by four the number of sensors deployed, the resolution is multiplied by 5.2 at a distance of 6 m and 11.5 at a distance of 12 m. This observation validates the chosen strategy of using low-resolution sensors to design photogrammetric models. Suppose the model’s resolution obtained with the minimum number of sensors is insufficient. In that case, it will be possible to increase it by adding additional sensors, with, of course, the disadvantage of increasing the cost and energy consumption of the whole. A compromise must therefore be found between these different parameters.

## 5. Sensors Energy Balance

As previously explained, monitoring a study area for photogrammetric reconstruction requires the multiplication of sensors, and to have an autonomous system for long-term monitoring (from 1 month to 1 year), it is necessary to evaluate the energy consumption of the system in different scenarios to optimize the life of the batteries. Our approach consisted of decomposing the image transmission through a network of communicating cameras into a series of elementary operations with measured energy consumption. The sum of the energy costs of the different scenarios allows us to evaluate the global energy needs and thus determine the optimal choices.

Traditionally, for wireless sensor networks, Zigbee or LoRa communication protocols are used. However, image transmission is a specific case where the amount of data transmitted is essential and, therefore, not well suited to these low-speed networks. The older Zigbee protocol has already been studied for image transmission, recommending compression [18] or the use of algorithms [19] to reduce the size of images and make them compatible with the transmission. In the context of our application, where the quality of the image matters and the use of algorithms would require over-consumption of the sensor node, these protocols are currently unsuitable.

Bluetooth and Wi-Fi are more suitable protocols for image transmission. Publication [20] shows that a solution to reduce power consumption is possible by using an event-driven communication mode and using the sleep mode between each transmission. Bluetooth has a slightly lower power consumption than WI-FI, but both protocols are regularly updated to allow new modes that can significantly reduce power consumption. In everyday use, Bluetooth is used more in peer-to-peer mode than in network mode. Here we want to easily increase the number of image sensors in our system, which leads us to use Wi-Fi despite the cost of a slight increase in power consumption.

For this feasibility study, we chose commercially and low-cost available development cards that do not require specific hardware design. More specifically, the card called ESP32-Cam was chosen to evaluate our scenarios and their energy consumption. This card contains a 32-bit microcontroller from Espressif: the ESP32, which has some low power functionality and Bluetooth and Wi-Fi connectivity. Moreover, a camera and an SD card reader are on the board and interface with the microcontroller.

All power measurements are performed in a standard way in the laboratory on the $V_{dd}$ pin used to power the microcontroller board. A two-step procedure was used to ensure that all current components were taken into account. The consumption measurements are made with stabilized power supplies. First, each measurement is validated by acquisition with a wide bandwidth (>100 MHz) thanks to an oscilloscope measuring the voltage at the terminals of a resistor placed between the laboratory power supply and the $V_{dd}$ pin of the microcontroller board. The objective of this first step is to ensure that all possible current peaks are considered. For a second time, a precise measurement of the current value is made with a precision amperemeter (DMM7510) in place of the resistor.

### 5.1. Initialization Energy Consumption

To characterize the minimal energy consumed by the chosen hardware, we have defined a standard initialization operation that will be present in our application. Whatever the feedback scenario, the boot operation of the card and the camera configuration do not change, and Figure 4 represents the current drained during the two phases.

Due to the complex architecture of the ESP32 and the many peripherals present, the card’s boot takes time, but the power consumption is controlled and remains below 80 mA, and we can see measured current in Figure 4a. These current measurements allow quantifying the energy for each operation, presented in Table 4.

The initialization energy $E_{init}$ for our system is then given by:(6)$$E_{init}=E_{Boot}+E_{CamConfig}=809.7\;\mathrm{mJ}$$

Whatever the image transmission scenario, the initial energy cost is fixed and determined only by the choice of the acquisition device and, to a lesser extent, by optimizing the start-up code. In our standard configuration, this energy cost is close to one joule per deployed sensor.

### 5.2. Energy for the Photo Shooting

Following the same procedure as before, the current consumption of a photo shoot is measured, and then the energy *E*${}_{shoot}$ consumed by this operation is calculated.

Concerning the photo shooting shown in Figure 5, the development card was again used in its most standard mode, with the exposure time set to automatic. In this configuration, the current measurement was repeated to take a picture of a plain background in different light conditions. To estimate the energy presented, we used the following formula *E* = V*<I>*Δt, where *E* stands for energy (J), V for voltage (3.3 V), <I> for average current (A) and Δt for time (s).

The results obtained, presented in Table 5, show that the energy consumed by the image capture is independent of the brightness.

### 5.3. Scenario Dependent Energy Consumption

The total energy balance depends on the scenario adopted for data transmission. In our study, we have chosen to transmit the photos from the sensors deployed at the edge of the study area through a Wi-Fi connection. Wi-Fi has the advantage of having a modular bit rate that can reach essential values under the needs of the image transmission. Nevertheless, this type of network is often absent from areas of interest for environmental photogrammetry, and its maximum range of about 250 m outdoors does not allow to follow the behavior of an area at a distance. The choice was therefore made to use a transportable relay allowing the sensor nodes to connect to the Wi-Fi, which will centralize the images before transmitting them on another long-range network such as GSM, as shown in Figure 6.

The daily energy cost of a mobile transmission has already been studied in the context of a sensor network [21,22]. It can be divided down into the cost of access to the network, estimated at around 190 joules per day ([22]), and the cost of data transmission, which for a 400 kB photo can be estimated at 21–23 joules according to the reference [21,22]. The long-distance transmission between our Wi-Fi relay and the server has a significant energy cost. Nevertheless, this single communication allows us to send all the photos, and only one deployment is necessary to cover a study area. The presence of a single GSM relay for all the sensors means that it can be powered by a solar panel at a lower cost [23].

In this study, we will focus on the image capture nodes, which should be as autonomous as possible. Once again, the energy balance of the system depends on the chosen scenario:Either the pictures are taken and stored on the SD card of each sensor to be transmitted to the relay by packet at a regular interval.Or after the photos are taken, they are transmitted directly to the relay.

The second scenario requires multiple daily Wi-Fi connections, thus generating a cost for the relay and the sensor. In contrast, the first minimizes the number of transmissions but requires an energy cost for the data storage. Adding a backup to the image sensor nodes is of interest when recovering deployed equipment and provides security in the event of a data transmission failure. That is why in the next section, we will estimate the energetic cost of this security and ensure it will not have a significant impact on the life of the system.

#### 5.3.1. Saving Photo and Transmitting

To determine the energy for both scenarios, the power consumption for saving an image to an SD card and transmitting the same image via Wi-Fi was measured. The results of the measurements are given in Figure 7. However, for each of them, the energy of the operation is related to the image size (thus to the amount of data saved or transmitted).

These measurements make it possible to calculate the energy of these operations, presented in Table 6.

In addition to this energy consumption, the configuration of the Wi-Fi, whatever the transmission that will follow, has an energy cost measured on our card at 0.23 J. It is necessary to do this configuration only one time per transmission session. In a session, N images can be sent until the code in the microcontroller ends the Wi-Fi connection or when an external event ( battery depleted, for example) shuts down the sensors. Putting the sensor in a deep sleep mode also ends the session. However, this energy remains low compared to the need for transmission or backup. Figure 8 shows the measurements made on these operations, and it can be seen that saving a photo consumes more than half the energy needed to transmit it, whatever the size of the photo tested. This cost is therefore significant for a redundancy system, which led us to eliminate this operation in the chosen scenario.

For the Wi-Fi transmission operation, the measurements also show excellent linearity between the energy drained and the size of the image. In detail, it can be seen that the average current consumed is constant during these operations. However, working on a larger image size increases the time taken by the operation proportionally. In the end, it is possible to estimate the energy cost of these operations by: (7)$$E_{Wi-Fi}=E_{ConfigWi-Fi}+\left(0.005*\left(ImageSizeinkB\right)-0.0594\right)*N$$
where $E_{ConfigWi-Fi}$ = 0.23 J, *N* is the number of Photos, and the 0.005 coefficient is established thanks to the measurement presented in Figure 8 and is specific to this microcontroller board.

#### 5.3.2. Wi-Fi Reception of Photo

In keeping with the idea of using commercially available devices, we used a Raspberry Pi (RPi) to receive the Wi-Fi data and re-transmit it to the mobile network. If, as explained above, there are already multiple studies on the energy balance of a mobile link and the powering of the Raspberry by solar panel, we have evaluated the additional energy cost necessary for deploying this multi-sensor photogrammetry system. Energy consumed by the multi-sensor system can be calculated as the sum of the energy related to the activation of the Wi-Fi on the device and that related to the reception of each photo, where n is the number of photos.
(8)$$E_{RPi}=E_{Wi-Fiactived}+N*E_{Photorx}$$

The energy related to the reception of a photo is obtained by measuring the consumption during the reception. In order to isolate the cost of reception from the consumption related to Wi-Fi activation, the current consumed by the RPi before the reception is subtracted from the calculation. Similarly, the standard consumption of the Raspberry Pi without Wi-Fi is subtracted from the energy related to Wi-Fi activation. These specificities are presented in Table 7.
(9)$$E_{Wi-Fiactived}=\left(<I{>}_{Wi-FiON}-<I{>}_{Wi-FiOFF}\right)*V_{supply}*T_{ON}$$
where $T_{ON}$ is the duration of RPi’s activity.

Considering that the RPi can operate on a solar panel in a GSM transmission mode, the addition of the photogrammetry system adds a low energy cost for Wi-Fi activation. However, the cost per photo will have to be controlled in the implementation not to limit the system’s autonomy.

## 6. Implementation and Associated Energy Costs

As the energy cost of SD card backup was considered too high, this solution was not retained in our scenario. The sensor program can therefore be presented in the form of the flowchart in Figure 9.

Communication with the RPi server allows the ESP32-Cam’s sleep time to be set. Figure 10 shows the power consumption of a complete cycle with two photos taken. The duration of the sleep mode was 5 min on this measurement.

Depending on the application, it is possible to increase the autonomy significantly, as shown in Figure 10, the deep sleep mode was measured on the board between two activations at 5 mA. It is also interesting to note that the energy required for the first boot is equivalent to the wake-up after the deep sleep mode. The equivalence is due to the use of a sleep mode where most devices are turned off, and the memory is not maintained, which imposes a new configuration cycle when waking up. It is then possible to consider the time spent in deep sleep as the time of activity minus the time T${}_{active}$ needed for picture taking and its transmission. This time has been measured on several configurations for an average value of 6 s (depending on the image size, the value is given for 400 kB). The energy of the system can be divided into two parts: the energy spent when the sensor is active and the one in a deep sleep according to the following equations: (10)$$E_{DeepSleep}=\left(0.005*3.3\right)*\left(T_{ON}-K*Tactive\right)$$
(11)$$E_{Active}=\left(E_{init}+E_{photoshoot}+E_{Wi-Fi}\right)*K$$
where *K* is the number of sensor Wakes-up. Considering a battery type LiFePO4 with a capacity of 3200 mAh for a nominal voltage of 3.3 V, it is possible to evaluate the theoretical autonomy of each image sensor (without battery self-discharge) using the previous calculation. The calculation result is presented in Figure 11 and shows the capacity of the sensor to hold 1.5 years with 24 photos per day. However, this calculation at 24 photos per day is very much oversized and corresponds to one photo per half hour in the daytime period. The events to be monitored along a river rarely (during floods) require such image density. However, it makes it possible to estimate the system’s autonomy for the worst case.

For the image information to be effectively transmitted, it is also necessary to calculate the energy requirement of the relay (RPi stands for Raspberry pi) to cover the same operating time as the image sensors. In our photogrammetric case study, a minimum of 17 sensors must be deployed. For each shot, the sensor will receive 17 photos. We consider taking 24 shots each day with a 400 kB photo size. With these parameters, Table 8 shows 502 days of autonomy for ESP32-Cam sensors.

In our scenario, the absence of standby or sleep mode for the Raspberry Pi leads to very high energy consumption during operation.

The addition of a Wi-Fi sensor network as presented to a GSM relay station based on an RPi can be quantified using columns 2 and 3. It is, therefore, 9.636 MJ for 502 days. The existence of a commercial solar kit allowing the Raspberry to be autonomous allows to power this system. The total energy of the relay station to consider when sizing the system is the sum of the four columns, which is about 89 MJ. Table 9 represents these needs reduced to one day in unit W.h.

The site [24] allows the calculation of the solar power available according to the geographical area of deployment of a self-powered solar system. This tool makes it possible to estimate the solar panel and battery system to provide the 49.25 W.H per day. A configuration in France on the edge of the Allier river leads to a solution comprising a 60 Wp (about 0.5 m${}^{2}$ of area) solar panel and a 90 W.H battery. The Wi-Fi range makes it easy to position the relay in a sunny area. However, this dimensioning, designed to operate even in winter with little light, is very much oversized in the summer.

## 7. Conclusions

We presented the photogrammetric technique to get a 3D rendering from 2D images. Although it usually performs the process with high-definition images, we have demonstrated the possibility of obtaining a 3D rendering with low-definition images.

Moreover, we justified the efficiency of the proposed method by its capacity to provide information during a major event on the river, allowing a fine analysis of its effects. In the second step, we discussed the energy efficiency of a scenario for image transmission. Besides, we described how the size of the images could influence energy consumption. Finally, through the article, it was possible to justify the trade-off between the resolution [25] (significant for the quality of the 3D rendering), the power consumption, and the cost of the module. In the worst case, the project’s feasibility is demonstrated with the autonomy of image sensors for more than one year. However, there are still investigations to be carried out to reduce the power consumption of the Wi-Fi to GSM relay. Following the conception of the system in the general case, we must apply it to two precise field configurations. First, on a small scale (a few m${}^{2}$), the monitoring of a patch of vegetation and its influence on sediment trapping will be carried out on an annual cycle. It could depend on the meteorological conditions and the configuration of the vegetation (presence or absence of leaves depending on the season) with a slow rate of shooting (one image per day). On a larger scale (a few tens of m${}^{2}$), the monitoring of dead wood accumulation areas (wood jams) is also possible, within this case monitoring only during flood period (2 months in the year) but with a higher rate of image taking (1 image per hour).

## Figures and Tables

**Figure 1 sensors-22-07617-f001:**
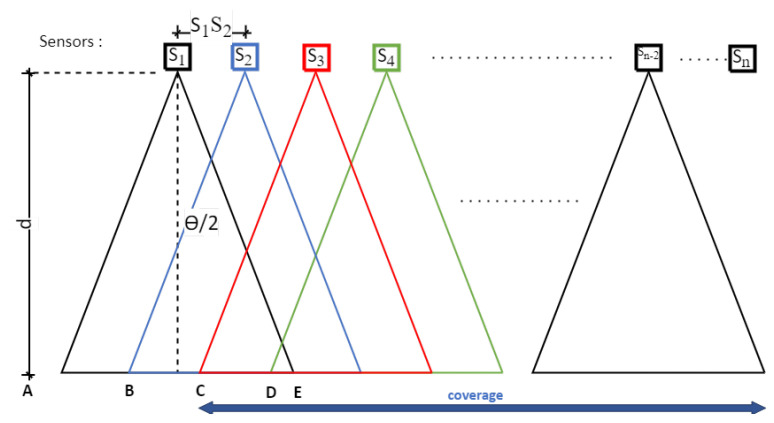
Sketch of sensor positioning respecting the visibility of a point on at least 3 cameras.

**Figure 2 sensors-22-07617-f002:**
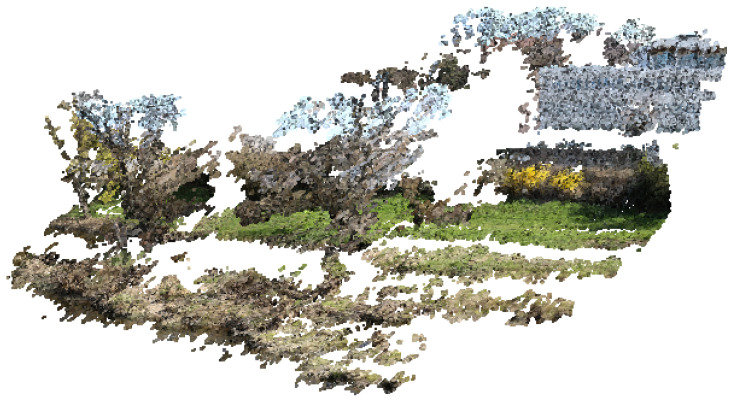
Overview of the 3D reconstruction from the 1600 × 1200 resolution.

**Figure 3 sensors-22-07617-f003:**
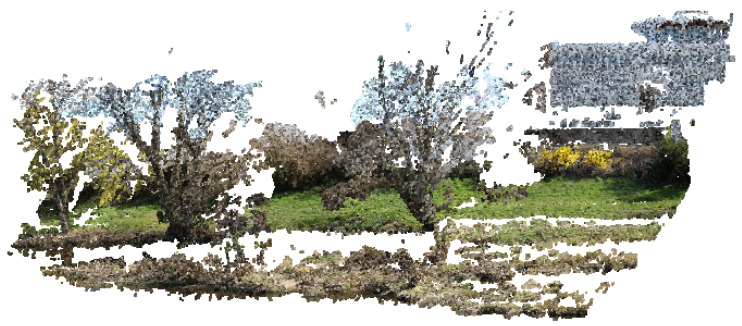
Overview of the 3D reconstruction from the 3200 × 2400 resolution.

**Figure 4 sensors-22-07617-f004:**
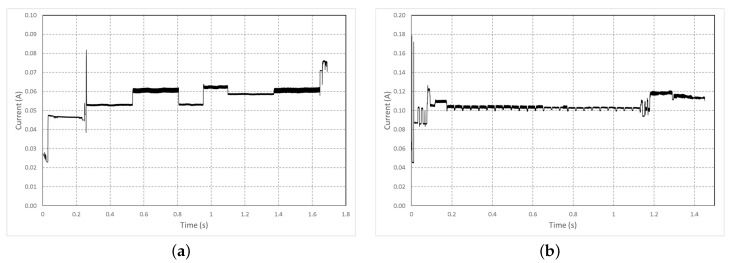
Measured current (**a**) during the boot sequence. (**b**) during the camera configuration.

**Figure 5 sensors-22-07617-f005:**
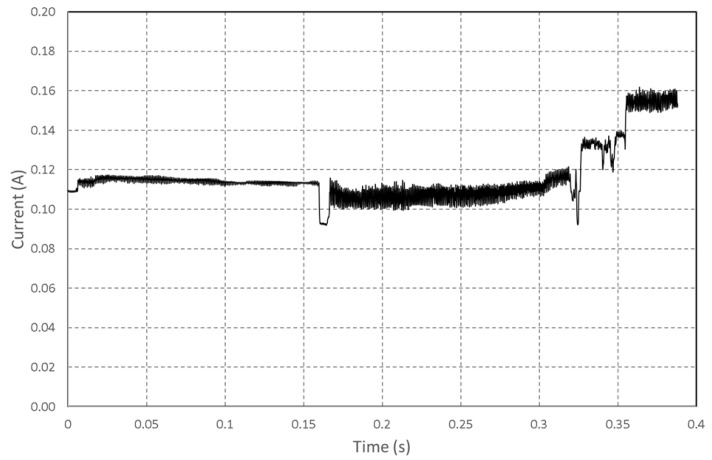
Measured current during shooting.

**Figure 6 sensors-22-07617-f006:**
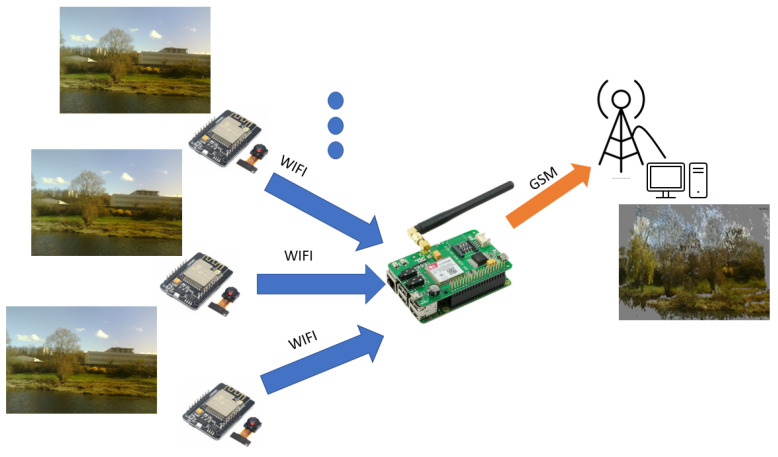
Description of the sensors network. Each image sensor communicates using Wi-Fi to the relay which transmits the images by GSM to the processing unit.

**Figure 7 sensors-22-07617-f007:**
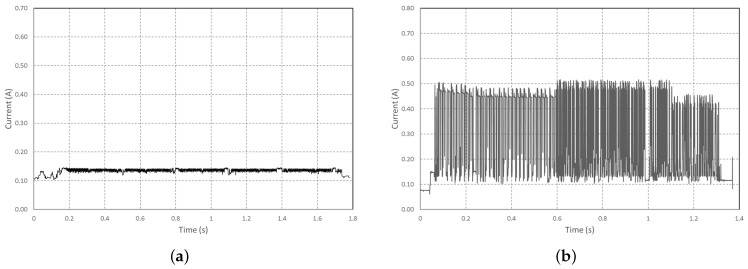
Measured current (**a**) during image saving on SD card. (**b**) during the Wi-Fi image transmission. Both for 270kB image size.

**Figure 8 sensors-22-07617-f008:**
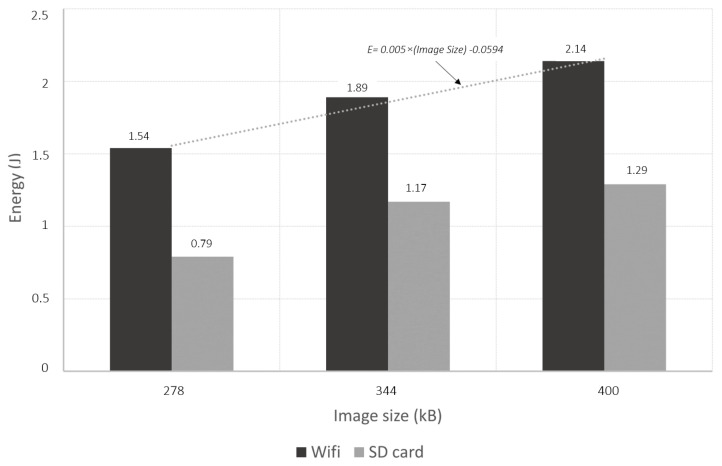
Energy consumed for Wi-Fi transmission and SD card recording on different image sizes.

**Figure 9 sensors-22-07617-f009:**
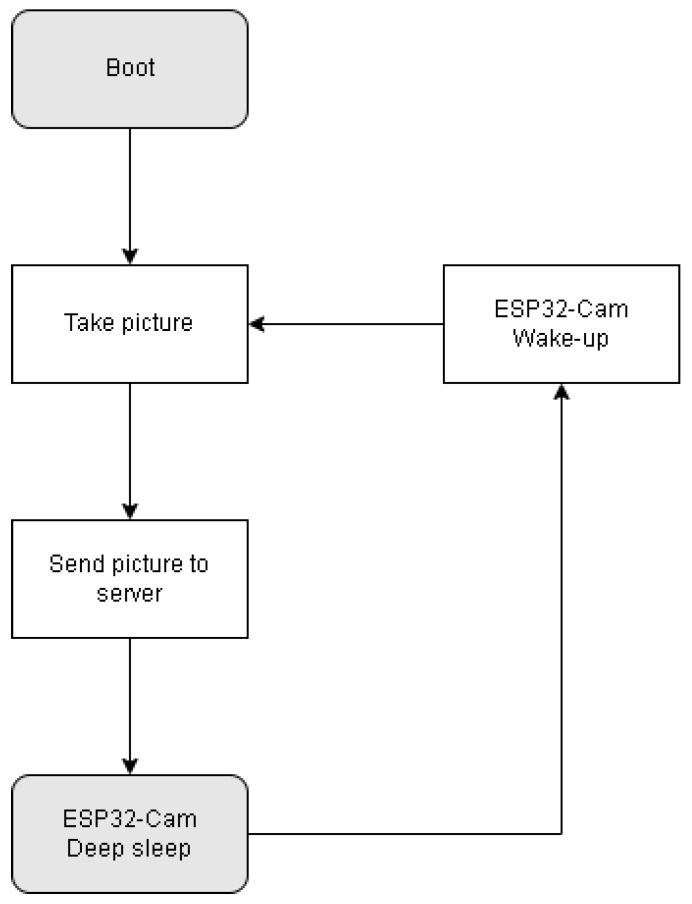
Simplified ESP32-Cam program structure.

**Figure 10 sensors-22-07617-f010:**
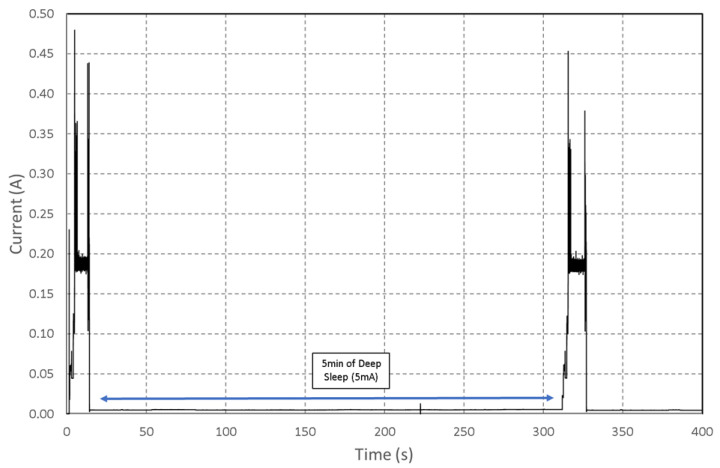
Measured of current consumption of the ESP32-Cam during a complete cycle with 2 photos taken.

**Figure 11 sensors-22-07617-f011:**
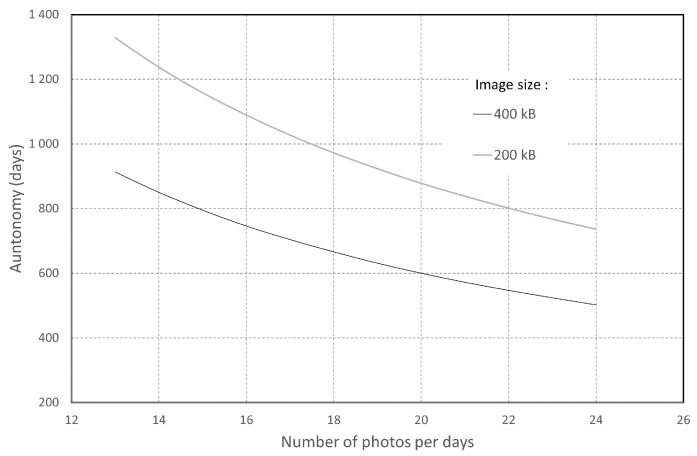
Estimation of the autonomy according to the number of pictures taken for 2 different image sizes (200 and 400 kB).

**Table 1 sensors-22-07617-t001:** The number of points obtained for the 3D model according to both image size (expressed in pixel) and image file size (expressed in MB). This test was carried out along the river, with 11 images taken at a distance of 3 m between each other.

Image Size (Pixel)	File Size (MB)	Number of Points of the 3D Model	Point Difference between Two Consecutive Resolutions (%)
4800 × 3600	4.8	26,115,500	0
4000 × 3000	3.6	18,756,125	−28
3200 × 2400	2.6	12,804,662	−32
2400 × 1800	1.6	7,234,022	−43
2000 × 1500	1.2	5,381,860	−25
1600 × 1200	0.816	3,659,032	−32
1200 × 900	0.480	1,651,659	−54
800 × 600	0.228	968,741	−41
400 × 300	0.077	0	−100

**Table 2 sensors-22-07617-t002:** The number of points obtained for the 3D model and point density according to both the distance between sensors and the number of sensors used. This test was carried out at a distance of 6 m from the investigated area.

Distance between Sensors (m)	Number of Sensors	Number of Points of the 3D Model	Point Density (pts/m${}^{2}$)
2	17	1,223,286	13,154
1.5	26	3,390,672	36,459
1	34	4,245,693	45,653
0.5	68	6,405,669	68,878

**Table 3 sensors-22-07617-t003:** The number of points obtained for the 3D model and point density according to both the distance between sensors and the number of sensors used. This test was carried out at a distance of 12 m from the investigated area.

Distance between Sensors (m)	Number of Sensors	Number of Points of the 3D Model	Point Density (pts/m${}^{2}$)
4	10	256,989	2763
3	15	294,080	3162
2	20	750,986	8075
1	40	2,951,147	31,733

**Table 4 sensors-22-07617-t004:** Energy.

Operation	Time (s)	Average Current (mA)	Energy (mJ)
Boot	1.66	56.1	307.3
Camera configuration	1.45	105	502.4

**Table 5 sensors-22-07617-t005:** Energy vs. luminosity.

Operation	Time (s)	Average Current (mA)	Energy (mJ)	luminosity (Lux)
Photo shooting	0.33	102.9	112	280
Photo shooting	0.33	102.7	111.8	500
Photo shooting	0.33	102.4	111.5	780

**Table 6 sensors-22-07617-t006:** Energy for image save or transmission.

Operation	Time(s)	Average Current (mA)	Energy (J)
Wi-Fi Transmission	1.37	290	1.31
SD card record	1.78	134	0.79

**Table 7 sensors-22-07617-t007:** RaspberryPi (RPi) Energy consumption.

Operation	Time (s)	Average Current (mA)	Energy (J)
RPi Boot	58.25	492	143.3
RPi without Wi-Fi	When ON	367	1.835 × T${}_{ON}$
RPi with Wi-Fi	When ON	411	0.22 × T${}_{ON}$
Photo reception	3.1 to 4	444.5	0.36 to 0.47

**Table 8 sensors-22-07617-t008:** Raspberry Pi (RPi) Energy consumption.

E for 17 Photos (J)	E for 24 Shot during 502 Days (J)	E RPi for Wi-Fi (J)	E for RPi Boot (J)	E for RPi Standard Activity during 502 Days (J)
0.47 × 17 = 7.99	7.99 × 24 × 502 = 96.3 K	9.54 M	143.3	79.59 M

**Table 9 sensors-22-07617-t009:** Daily energy requirement.

Power Needed by Complete Relay Station (W.H/day)	Power Needed for Wi-Fi Connection to Sensors Network (W.H/day)
49.25	5.33

## Data Availability

Not applicable.

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
