# Peer review of "Low Power Environmental Image Sensors for Remote Photogrammetry"

_sensors, 2022, doi:10.3390/s22197617_

Round 1

Reviewer 1 Report

Summary:

-          This manuscript describes and evaluates a low-cost, low-energy approach for long-term monitoring of remote sites to support improved understanding of morphological changes in river systems over time.  Key components of the work are enabling data collection without human intervention (for operating cameras or replacing batteries) and assessing the quality of data analytics that can be achieved with the proposed sensor network.  The motivation for the work is clearly laid out, and the scientific benefit of the work is well described, however gaps remain in putting the analyses completed in context of the scientific question as well as providing sufficient information about analyses conducted for the reader to properly evaluate the results.

Abstract: 

-          In place of “for the dynamics of a river network” the reader would benefit from a more specific description on exactly what is being measured (e.g., flow, level, detritus, bank location, etc.)

-          In place of this sentence “The ESP32-Cam has shown its ability to meet the different requirements of our specifications” it would be helpful to include a list of the actual specifications (e.g., resolution, cost, memory, etc.). 

-          Similarly it would be helpful for the sentences discussing the energy analysis to include the final energy consumption number (mWh/day or similar).

Line 17:  Please provide a few specific examples of the “many contexts”

Figure 1:  For optimal coverage why would the black and green point not converge at Point D?  Can the authors please clarify in the text/figure caption whether the goal of the approach is to have cameras along a straight line (as implied in Figure 1) or whether this line of sensors would, for instance, follow a curving river bank?  If the latter can the authors please provide more detail on whether the calculations provided are conservative or may result in data gaps?  Note:  in general a more detailed caption on Figure 1 would be helpful for the reader, with a minimum of defining all variables used in the figure.

Table 1:  Three comments.  (1) Is the number of 3D model points not dependent on the distance from the river in addition to the spacing of the images?  Can this information be added into the table?  (2) The information in this table would be more valuable in the context of some objective evaluation of how many features exist in the landscape captured in these 11 images.  That is – in an expert evaluation, how many points are actually needed in the 3D model to understand the landscape?  While the authors offer some tradeoff analysis (Line 102) it does not provide any reference back to the original problem statement to ensure that the selection of camera is indeed still producing a meaningful model.  (3) Are there commercial camera options corresponding to each of the lines in the table?  If so, can that information be added as another column, potentially also with cost and power consumption as comparative metrics?  If not, can the authors provide a motivation for why these specific resolutions were selected?

Line 110:  Please provide more explanation for how the surface area of 93m^2 is determined.

Tables 2-3:  Please provide details on the software used to identify 3D model points from the photographs.  Please comment on the average size of features in the landscape, especially in reference to the surface area that would be represented by one pixel in each of the two scenarios.  For instance, does this area typically have boulders of size 1-2m diameter?  Scree of size <1cm diameter?  Trees?  Other features?  An additional figure showing images of the site and how they are overlain would be of significant value to the reader in assessing the approach in the context of the science question.

Line 127:  Here the authors define a metric of success (1 point per cm^2) relative to the scientific goal.  It would be helpful to have this discussion and target success metrics presented much earlier in the manuscript as context for the reader to understand all analyses related to resolution and pixel size.

Line 137-138:  The message the authors are repeating here does not clearly come out in the early part of the manuscript – per previous comment, it would be helpful to increase level of detail in the early part of the manuscript to provide this context for the reader.

Lines 138-146:  The conclusion stated here is obvious (more sensors means more data).  It would be helpful for the analysis to focus more primarily on how to define the needed number of sensors as a function of the landscape to be studied.

Line 168:  Please provide an explanation for how current was measured.  What is the setup, instrumentation, sampling rate?  How have the authors ensured that there are no high frequency spikes that are being missed (e.g., as would be due to use of a flash)?  How was it determined when some step was “finished”?  Especially for Wifi transmission, please justify how mean current is appropriate for calculating power consumption?

Table 4, Table 5, and all energy calculations:

-          Please present energy calculations in mWh for easier comparison to commercial battery capacities. 

-          For these calculations is the supply voltage assumed or measured? 

-          Are these calculations taking into account any energy losses in generating a regulated 3.3V supply, e.g., losses in DC/DC converters from a 5V USB supply or similar?

-          Is each measurement taken only once?  A statistically representative sample with mean and standard deviation is needed to evaluate the results.

4.1 Energy costs – can the authors please provide more detail surrounding assumptions in the calculations provided?  Are these only initial boot or, given that (Line 262) the manuscript reports that boot-from-sleep has same power consumption, are these data presented here an average of both scenarios?  Discussion of sleep mode waking should also be discussed here for clarity.  Also given that sleep mode is later considered for minimization of energy consumption, why is the measurement or manufacturer reported value for sleep mode power consumption not included in the manuscript?

Line 185:  Variables are not defined.

Table 5:  Where was lux measurement taken in proximity to the camera?

Section 4.2:  Is energy requirement for recording data to SD card included in energy requirement for capturing an image?  If not please explain the definition of the end point of this measurement – the data are saved in RAM?

Section 4.3:  Can the authors provide more discussion of options that were available for transmission and were compared?  For instance, the devices have Bluetooth and a Bluetooth mesh network might have been considered, or inclusion of a GSM module for each sensor may have been an option.  It may also be important to discuss the implications of Wifi being limited to line-of-sight, depending on the landscape of the expected field sites.

Line 212-213:  This statement is valid is not logically well connected to the adjacent narrative in the manuscript.

Figure 5:  Is the SD card initialized in the boot-up of the camera?  If not, where is that energy consumption accounted in the work flow?

Line 221:  How frequently does the Wifi configuration happen?  Once at startup or once per transmission?  How many “maintenance” type transmissions happen to keep the connection to the central node live?

Section 4 general comment:  references made to “solar panels” require more detail in description as, theoretically, anything could be powered simply by increasing the size of the solar panel.  The problem statement should have some constraint in terms of size of solar panel (in W), availability of sunlight (e.g., on shortest day of the year or average day of the year), budget for solar panel and battery, or other metrics which provide the appropriate feasibility analysis constraints.

Line 271:  Why has this battery size been selected as opposed to a battery of any other size?  Cost optimization?  Commercial pairing with the selected device?

Line 274:  Can the authors please comment on whether 24 photos per day is an appropriate amount needed to achieve the scientific aims of understanding changes in the river system?  How many photos per day are actually necessary?  What would be the lifetime of the system (battery) at that photo rate?

Lines 285-287:  The calculations here are not clear.  What is the suggested battery powering?  Is this battery being recharged by a solar panel?  How is this integrating with the suggested solar system?

Suggestions for grammar revisions:

Line 23-25:  “the shooting of”, remove “sector”

Line 42:  after comma should be “for which the 3D monitoring was conducted”

Line 96:  unclear what is meant by “weight” – file size?

Line 171:  shortened reference to the figure would make this sentence more readable

Line 184:  unclear what is meant by “presented” in this context

Line 192:  unclear what is meant by “important” in this context

Line 234:  unclear if “card” refers to microcontroller card, wifi card, or something else

Line 246:  Acronym RPI not defined in the text

Author Response

Answer is given on attached word document.

Reviewer 2 Report

The paper presents an important topic especially with the widespread and adoption of IoT  in many applications and scenarios. However, the paper needs extensive revision in order to make it publishable in this high quality journal.

Formatting points

In general the paper is well written and has good formatting. However, the below are some minor points that need improvement:

-          The authors some time use WIFI or WiFi or wifi (in the conclusion ). So please be consistent. Check also other acronyms for the same point.

-          The authors do not use the math formatting for the math symbols inside the text (e.g. n in line 78,  line 185, and many others)

-          Line 75 ( view ()) (empty brackets, please check

Technical feedback

1.The introduction section is too short, I suggest expanding it and also add a literature review section or part to it.

2.Please clarify how your work is different from related work. What is your contribution compared with other related work, which can be elaborated using a table of comparison?

3.As the authors are concerned about energy consumption, why they did not investigate the potential adoption of Zigbee or LoRa (as mentioned in the below references).

4.WiFi is known of having high energy consumption and low range, while Zigbee for example, has low energy and longer range, The same applies to LoRa. Please explain why not using these as alternatives to WiFi.

Supporting reference

Zinonos, Z., Gkelios, S., Khalifeh, A.F., Hadjimitsis, D.G., Boutalis, Y.S. and Chatzichristofis, S.A., 2021. Grape Leaf Diseases Identification System Using Convolutional Neural Networks and LoRa Technology. IEEE Access10, pp.122-133.

5.Why not also proposing to use the solar panels to the end point and not only at the relay point?

6.Please provide more explanation on how you measure/estimate  the energy consumption, what measurement tools did you use. I suggest adding a separate section about that, since it is very important.

7. Did you check if you can compress the images before transmission, if yes, please explain more, if not, please explain why.

Reviewer 3 Report

This is a prospective study that investigates the Low power of environmental image sensors for remote photogrammetry. The following are my comments and critique:

  1. There is no solid justification for choosing 1600X1200 as an appropriate image size for monitoring. How about the justification for the quality of the point cloud for feature extraction? Is there any trade-off? (Line 102) 

  2. One of the major aspects which need to be considered in this study is the trade-off between the energy consumption aspect and the data quality aspect for environmental monitoring. Is there any preliminary sensitivity analysis you have done that can clearly justify your conclusion?

  3. You need to define what RPI stands for in an appropriate way.

  4. In Eq. 7 how do you define the 0.005 coefficient? Is it part of the manufacturer's specification? You need clearly define where that number is from.

  5. In Figure 4, you presented the GSM technology for data transmitting to the server but it was never investigated in terms of energy consumption. I believe that is a part of data transmission which needs to be investigated. How about if you use other existing technology including LoRaWAN?

Round 2

Reviewer 1 Report

The abstract has not been improved to include key quantitative details from the paper.

The visual in Figures 2-3 is very helpful but the quantitative method may rely on a level of data for which the difference would not be visible at this size/print resolution.  This discussion needs to be founded in the needs of the numerical method for relevance.

Added text requires copy editing for English language correctness.

Explanation of how current measurements were made is still lacking, i.e., it is not possible to replicate the setup because there is no circuit diagram, instrument make/model, etc.

Reviewer 2 Report

Thanks, i still foudn the answer of (Please provide more explanation on how you measure/estimate the energy consumption, what measurement tools did you use. I suggest adding a separate section about that, since it is very important.) is not clear and not detailed. Please provide a more clear and detailed answers, so people can follow your methodology if they want to measure the energy consumption, and add it to the paper).

Regarding the solar panel integration, I do not mean to use it as a replacement, but it can be added to charge the battery, please elaborate on that possibility on the paper.
